# Analysis of Adverse Effects of COVID-19 Vaccines Experienced by Healthcare Workers at Guizhou Provincial Staff Hospital, China

**DOI:** 10.3390/vaccines10091449

**Published:** 2022-09-02

**Authors:** Yunhua Wei, Yan Wang, Lin Liu, Yan Zha, Yuqi Yang, Yuanlin Wang, Neil Roberts, Yaying Li

**Affiliations:** 1Department of Nuclear Medicine, Guizhou Provincial People’s Hospital, Affiliated Hospital of Guizhou University, Guiyang 550002, China; 2Department of Obstetrics and Gynecology, Guizhou Provincial Staff Hospital, Huaxi Branch Affiliated to Guizhou Provincial People’s Hospital, Guiyang 550003, China; 3Department of Respiratory and Critical Care Medicine, Guizhou Provincial People’s Hospital, Affiliated Hospital of Guizhou University, Guiyang 550002, China; 4Department of Nephrology, Guizhou Provincial People’s Hospital, Affiliated Hospital of Guizhou University, Guiyang 550002, China; 5Department of Urology, Guizhou Provincial People’s Hospital, Affiliated Hospital of Guizhou University, Guiyang 550002, China; 6School of Clinical Sciences, The Queen’s Medical Research Institute (QMRI), University of Edinburgh, Edinburgh EH8 9YL, UK

**Keywords:** inactivated COVID-19 vaccine (Vero cell), recombinant novel coronavirus vaccine (CHO cell), adenovirus type-5 (Ad5) vectored COVID-19 vaccine, healthcare workers, adverse events following immunization

## Abstract

Objective: A retrospective survey was conducted of adverse events following immunization (AEFI) experienced by health care workers (HCWs) in a relatively remote ethnic region in southwest China (Guizhou Province) who received COVID-19 vaccines. Methods: From 18 January 2021 to 21 January 2022, all HCWs of Guizhou Provincial Staff Hospital, China, who received at least one dose of inactivated COVID-19 vaccine (Vero cell), recombinant novel coronavirus vaccine (CHO cell), or one dose of adenovirus type-5 (Ad5) vectored COVID-19 vaccine were asked to complete a self-report questionnaire to provide information on any adverse events that may have occurred in the first 3 days after injection. The frequency of AEFI corresponding to the three types of vaccines were compared and the potential risks of AEFI due to the three different vaccines were predicted by multivariate logistic regression analysis. Results: Of the 904 HCWs who completed the survey, the rates of AEFI were 10.1% (80/794) due to Vero cell, 16.3% (13/80) due to CHO cell, and 46.67% (14/30) due to Ad5 vectored vaccines, and the rates were significantly different (χ^2^ = 38.7, *p* < 001) between the three vaccines. Multivariate logistic regression models predict that (1) compared to the Ad 5 vectored group, the risk of AEFI occurrence in the Vero cell group was reduced by about 85.9% (OR = 0.141, 95% CI: 0.065–0.306, *p* < 0.001) and in the CHO cell group by about 72.1% (OR = 0.279, 95% CI: 0.107–0.723, *p* = 0.009), (2) the odds for women experiencing AEFI were about 2.1 (OR = 2.093, 95% CI: 1.171–3.742, *p* = 0.013) times as high as those of men, and (3) the risk of AEFI for HCWs with a Bachelor’s degree or above was about 2.2 (OR = 2.237, 95% CI: 1.434–3.489, *p* = 0.001) times higher than in HCWs who do not have a Bachelor’s degree. Conclusions: 1. The inactivated COVID-19 vaccine (Vero cell), recombinant novel coronavirus vaccine (CHO cell), and adenovirus type-5 (Ad5) vectored COVID-19 vaccine made in China are safe and relatively broad-spectrum. 2. The prevalence of AEFI is more common in women healthcare workers. 3. The risk of AEFI was higher in those with a Bachelor’s degree or above and may be related to the psychological and social effects triggered by the global COVID-19 pandemic.

## 1. Introduction

Following the discovery of the first cases of severe acute respiratory syndrome coronavirus 2 (SARS-CoV-2) in December 2019, vaccination against coronavirus disease 2019 (COVID-19) is considered the best way to limit the size and severity of the pandemic [1,2,3], and in China, the main types of vaccine are inactivated COVID-19 vaccine (Vero cell), recombinant novel coronavirus vaccine (CHO cell), and adenovirus type-5 (Ad5) vectored COVID-19 vaccine. The inactivated vaccine is widely used for the prevention and control of emerging infectious diseases, including influenza virus and poliovirus. Similarly, this mature technology has been applied to the COVID-19 vaccine [4]. The recombinant novel coronavirus vaccine (CHO cells) is a recombinant vaccine based on a prefusion-stabilized spike trimer of SARS-CoV-2. The spike protein was expressed in Chinese hamster ovary (CHO) cells, purified, and supplemented with an aluminum hydroxide adjuvant to improve immunogenicity [5]. The Ad5 vectored COVID-19 vaccine was developed by the Beijing Institute of Biotechnology (Beijing, China) and CanSino Biologics (Tianjin, China). The vaccine is a replication-defective Ad5 vectored vaccine expressing the spike glycoprotein of SARS-CoV-2 [6]. All three vaccines had good safety and tolerability in clinical trials. However, adverse events following immunization (AEFI) need to be monitored in different environments in order to provide clinicians and the public with further information on which to base decisions regarding immunization.

At the beginning of the epidemic of SARS-CoV-2 (2019–2020), the Guizhou Provincial Staff Hospital was designated by the local government as a special medical institution for the treatment of COVID-19 in Guizhou Province, and a total of 144 confirmed COVID-19 patients were admitted in 2020. Health care workers (HCWs) have worked at the frontline of the response to the new coronavirus disease 2019 (COVID-19), and have been at a higher risk of acquiring the disease and, subsequently, exposing patients and colleagues. They have thus received COVID-19 vaccines as soon as they become available. There are presently few reports on AEFI experienced by HCWs who received different types of COVID-19 vaccine in specific ethnic areas. In the present study, a retrospective analysis was performed for HCWs in a large district hospital in Guizhou Province, a relatively remote ethnic region in southwest China and the home of a population of predominantly Han, Miao, Buyi, Dong, Tujia, Yi, and Gelao ethnic groups. In particular, adverse events following immunization and related factors have been analyzed for HCWs in Guizhou Provincial Staff Hospital, who were immunized with at least one dose of the inactivated COVID-19 vaccine (Vero cell), recombinant novel coronavirus vaccine (CHO cell), and adenovirus type-5 (Ad5) vectored COVID-19 vaccine. Additionally, in our study, we hoped that more useful information would be obtained to enrich the post-COVID-19 vaccination data.

Among the publicly available research reports, there are none in which the risk factors for AEFI after the use of different types of COVID-19 vaccines in HCWs are reported. The present study used a self-report questionnaire made available to HCWs via a mobile phone app to determine the risk factors for AEFI following the use of three COVID-19 vaccines, and has provided information that is potentially useful for decision-making regarding strategies for COVID-19 vaccination.

## 2. Methods

### 2.1. Study Design and Population

A total of 1000 HCWs at Guizhou Provincial Staff Hospital who had received at least one dose of a COVID-19 vaccine between 18 January 2021 and 21 January 2022 were invited to complete a mobile phone-based questionnaire. The study was approved by the Institutional Review Board. Because the data were de-identified, the Institutional Review Board waived the necessity for written informed consent from each participant. Furthermore, no personally identifiable information was acquired and no human biospecimens were obtained.

### 2.2. Data Collection Procedure

The initial COVID-19 vaccination protocol in China was as follows: (i) inactivated vaccines (Vero cell) administered as two doses with an interval of 3 to 8 weeks, (ii) recombinant protein subunit vaccine administered as a single dose, or (iii) recombinant protein subunit vaccine (CHO cell) administered as two doses with an interval of 4 to 8 weeks. In addition, a COVID-19 vaccine booster is given to people over 18 years old 6 months after the initial COVID-19 vaccination. The dose of each vaccine was administered according to the manufacturer’s instructions.

The first phase of the vaccination program was primarily focused on HCWs nationwide, with exclusive vaccination sites in hospitals. In the Guizhou Provincial Staff Hospital, the HCWs who received one dose of the COVID-19 vaccine used a mobile phone app to report the symptoms of AEFI that occurred within 3 days of vaccination. Anonymity and confidentiality of participants were assured by not collecting personal identifying information. The questionnaire included requests for information regarding sex, age, ethnicity, level of education, a history of COVID-19 infection, wishes in respect of COVID-19 vaccination, history of allergies to the vaccine, types of vaccinations, doses of vaccinations, symptoms of AEFI within 3 days of vaccination, and whether medical attention was required after administration of the vaccine. If the answer to the last question was yes, then further information was requested with respect to whether the patient received out-patient or in-patient treatment and details of the treatment. HCWs who answered the entire questionnaire were divided into three groups, namely, the Vero cell group, CHO cell group, and Ad5 vectored group.

### 2.3. Inclusion Criteria

Inclusion criteria were HCWs who had received at least one dose of inactivated COVID-19 vaccine (Vero cell), recombinant novel coronavirus vaccine (CHO cell), or adenovirus type-5 (Ad5) vectored COVID-19 vaccine.

### 2.4. Exclusion Criteria

(1)Not an HCW;(2)People with a history of COVID-19 infection or vaccine allergies;(3)Women who were pregnant or breastfeeding;(4)People with severe chronic or immunocompromised diseases;(5)People aged < 18 years or >60 years.

### 2.5. Statistical Analysis

Statistical analysis was performed using GraphPad PRISM version 9.0.0 (GraphPad Software, CA, USA) and SPSS version 26 (IBM Corp., Armonk, NY, USA), and the results were considered significant for *p* < 0.05. Values of categorical variables were recorded as frequencies or percentages and were analyzed using the χ^2^ test or Fisher’s exact test. Summary descriptions were prepared regarding the baseline characteristics, whether an HCW required medical attention after administration of a vaccine together with vaccination reactions for the Vero cell group, CHO cell group, and Ad5 Vectored group. The *p*-value was adjusted by Bonferroni when performing a comparison of the χ^2^ test for multigroup rates. A multivariate logistic regression analysis was performed to determine the relative potential risk of AEFI, and the factors associated with AEFI, for the different vaccine groups. In addition, univariate analysis was performed to assess potential covariation (including sex, age (stratifying by age groups), ethnicity, level of education, and vaccine groups) and forward adjustments were subsequently made for statistically significant covariates in the multivariate logistic regression models. The odds ratio (OR) and 95% confidence interval (CI) were calculated for the regression model.

## 3. Results

Of the 1000 HCWs who received the questionnaire, four had a weak desire to get vaccinated. No HCWs had ever been infected with COVID-19. Finally, 904 completed a full response and were included in the study, of whom 73.33% (663/904) were women and 50.0% (452/904) had a Bachelor’s degree or above (Table 1). The Vero cell group vaccine was received by 87.83% (794/904), the CHO cell group vaccine by 8.85% (80/904), and the Ad5 vectored cell group vaccine by 3.32% (30/904) of participants. Table 1 also shows the distribution of ethnicity among the respondents who received the three types of COVID-19 vaccines, but it was not statistically significant (χ^2^ = 1.3, *p* = 0.969). Overall, 11.83% (107/904) of participants experienced an AEFI during the first 3 days after receiving the vaccine, and further details of the responses can be found in Table 2. Only, four people sought help from the out-patient provider, but none underwent any medical treatment and the symptoms of AEFI resolved spontaneously.

There were significant differences in the incidence of AEFI between the three groups (Figure 1), with values of 46.7%, 16.3%, and 10.1% in the Ad5 vectored, CHO cell, and Vero cell groups (χ^2^ = 38.7, *p* < 0.001), respectively, and with fever (26.7%), pain at the injection site (10%), and muscle pain/headache (13.3%) the most commonly reported AEFI, respectively (Figure 2). The results of the comparison between the three groups showed that the frequency of AEFI was significantly higher in the Ad5 vectored group compared with both the Vero cell (*p* < 0.05) and the CHO cell groups (*p* < 0.05) (Figure 1).

The potential risk of AEFI between the three vaccine groups and the factors associated with AEFI were subsequently computed using an adjusted multivariate logistic regression model, and the results are presented in Table 3. The incidence of AEFI was significantly higher in the Ad5 group than in the other two groups and univariate analysis subsequently showed that relative to the Ad5 vectored group, the risk of AEFI was reduced by about 87.2% (OR = 0.128, 95% CI: 0.060–0.272, *p* < 0.001) in the vs. Vero cell group and by about 77.8% (OR = 0.222, 95% CI: 0.087–0.563, *p* = 0.002) in the CHO cell group. The univariate analysis also showed that the risk of AEFI was about 2.4 (OR = 2.428, 95% CI: 1.377–4.280, *p* = 0.002) times higher in women compared with men and about 2.1 (OR = 2.055, 95% CI: 1.348–3.134, *p* = 0.001) times higher in participants with a Bachelor’s degree compared to those without. However, there were no significant differences in the risk of AEFI depending upon age group or ethnicity (*p* > 0.05).

Subsequent application of an adjusted multivariate logistic regression analysis showed that compared to the Ad5 vectored group, the risk of AEFI was reduced by approximately 85.9% (OR = 0.141, 95% CI: 0.065–0.306, *p* < 0.001) and 72.1% (OR = 0.279, 95% CI: 0.107–0.723, *p* = 0.009) in the Vero cell group and the CHO cell group, respectively. The risk of AEFI was approximately 2.1 (OR = 2.093, 95% CI: 1.171–3.742, *p* = 0.013) times higher in women compared to men, and approximately 2.2 (OR = 2.237, 95% CI: 1.434–3.489, *p* < 0.001) times higher in those with a Bachelor’s degree compared to those without.

## 4. Discussion

The incidence of AEFI in HCWs who had received one of different vaccines for protection against COVID-19 was investigated in HCWs in a large district hospital in Guizhou Province. In our study, the most common AEFI was fever, pain at the injection site, headache, or muscle pain, similar to other conventional vaccines, such as the whole-cell pertussis vaccine [7], human papillomavirus (HPV) vaccine [8], or influenza vaccine [9]. Severe adverse events were rare, and included hypersensitivity, facioplegia, urticaria, and anaphylactic shock [10]. The occurrence of adverse reactions varies in response to different antigen immunizations. Fever can occur in about 10% or more of vaccines [7]. Other mild systemic reactions (e.g., headache) are also common occurrences after vaccination. For example, after immunization with the bivalent HPV vaccine, the occurrence of fatigue and headache can be up to 33.0% and 30.0%, respectively [7]. However, the most common occurrence of headache/muscle pain was only 5.2% in our study.

Similar to other studies performed in China, none of the vaccines caused any serious or rare AEFI [6,11,12]. Vaccine-associated anaphylaxis is rare. Influenza vaccination has been associated with Guillain-Barré syndrome, but the absolute risk is very low (about 1–2 additional cases per million persons vaccinated) [9]. In addition, a phase 3 clinical trial of the recombinant novel coronavirus vaccine in the UK [13] and an international multicenter clinical trial of the adenovirus type-5 (Ad5) vectored COVID-19 vaccine [14] both showed that the COVID-19 vaccines are efficacious and safe in healthy adults aged 18 years and older. The international multicenter clinical trial study of the Ad5 vector vaccine has reported 24 serious adverse events, but none of the serious adverse events were assessed as vaccine related. The frequency of AEFI in the Ad5 vectored group was significantly higher than in the Vero cell group and CHO cell group, which is consistent with previous reports of both national and international studies [15]. Although the Ad5 vector vaccine is a novel COVID-19 vaccine, a clinical trial in China showed that the Ad5-vectored COVID-19 vaccine has a good safety profile, with only mild, transient adverse events related to vaccination and no serious adverse events [16]. An AEFI is any adverse medical event occurring after the immunization, which is not necessarily causally related to the vaccination. However, in clinical application, a certain probability of adverse effects happens by chance. In most cases, the exact mechanism of adverse reactions caused by vaccination is unknown, but it may be related to non-specific immune responses to components of the vaccine (e.g., adjuvants, stabilizers, or preservatives). In the present study, it is not possible to verify the cause of the AEFI, but the symptoms of AEFI reported were both transient and self-limiting, and serious adverse effects were rare, suggesting that all three vaccines are safe.

The observation that the prevalence of AEFI was more common in women than men is consistent with previous studies [17,18,19] and is attributed to immunological, hormonal, or genetic factors. In adults, women tend to exhibit a stronger immune system response to vaccines than men [20]. Specifically, women develop higher antibody responses and experience more adverse events following vaccination. For example, angiotensin-converting enzyme 2 (ACE2) is a functional receptor against SARS-CoV-2 [21,22]. This enhanced female immunoregenicity is thought to make women more resistant to infectious diseases, but instead, it also contributes to a higher incidence of women’s autoimmunity. The research found that the estrogen 17β-estradiol inhibits ACE2 activity, but androgen upregulates the activity of ACE2 [23]. Therefore, sex may be a critical factor in the design of future vaccine trials, and long-term monitoring of women with new vaccination is necessary. Personalized vaccines, customized to address sex-immune profile variation, may offer greater protection against both infectious as well as non-infectious targets. In addition to biological effects, in a cross-sectional study [24], the sensitivity of symptoms perceived might be high in HCWs with their higher level of education and experience in the healthcare field. Therefore, differences in education level may influence the response to vaccines. In addition, attention should be paid to the psychological well-being and social care of HCWs. Guizhou province is a relatively remote ethnic region with an economy and cultural assets that lag behind those of more developed cities. In January 2020, 144 patients with SARS-CoV-2 were admitted to Guizhou Provincial Staff Hospital. COVID-19 led to the beginning of a worldwide pandemic, and one of the groups most exposed to the virus and its psychosocial consequences is healthcare workers, due to their close proximity in caring for affected people. In fact, early research conducted on the Chinese population showed that a significant proportion of health care workers have depression symptoms (50.4%), anxiety (44.6%), insomnia (34%), and discomfort (71.5%) [25]. This evidence suggests that HCWs can be more vulnerable than others to the psychosocial effects of pandemics. Unpleasant emotions can cause the body to generate very high stress levels, which can disturb the functioning of the immune system, central nervous system, and endocrine system [26,27]. Therefore, to reduce the frequency of psychogenic responses, it is advised that psychological counseling management be included in the monitoring and management of vaccination in this population.

In a previous study, it was reported that increasing age is a significant risk factor for the incidence of AEFI following vaccination against COVID-19 [19]. However, in the present study, the three vaccines produced in China were found to be safe for HCWs of all ages. Ethnicity has also been reported to be a significant risk factor for AEFI, with people identifying as Asian more likely to report adverse effects. However, the reason for these differences remains unexplained [28]. The population of Guizhou Province in China includes members of 56 different ethnicities, of which 18 ethnicities, including Han, Miao, Buyi, Dong, Tujia, Yi, and Gelao, have lived there for several centuries [29]. Thus, a response regarding ethnicity was included in the questionnaire and used as a covariate in the exploratory multivariate logistic regression model. However, ethnicity was not a significant risk factor for the incidence of AEFI (*p* > 0.05), suggesting that the three vaccines produced in China may not exhibit a specific risk with respect to ethnicity and region. This is the first report on the correlation between Chinese ethnicity and the adverse reactions of COVID-19 vaccines. In the future it is likely that large digital cohort studies will be performed and provide more detailed information.

From late 2020 to the present day, the continuous emergence and evolution of Alpha, Beta, Gamma, Delta, and Omicron variants with different transmissibility and producing diseases of different severity has been a notable feature of the COVID-19 pandemic. There is compelling evidence suggesting that COVID-19 vaccines reduce the severity of infection and prevent death. Nevertheless, recent studies have shown that patients with COVID-19 may suffer from a series of sequelae, such as cardiovascular disease [30], sexual dysfunction in men [31], and brain atrophy [32], which seriously affect the quality of life of patients. Consequently, vaccination has played a key role in controlling the pandemic [33], and the work of HCWs has been crucially important. In 2020, Guizhou Provincial Staff Hospital was designated by the local government as a specialized medical institution for the treatment of SARS-CoV-2 patients. Very few HCWs refused vaccination in Guizhou Provincial Staff Hospital due to vaccine hesitancy. Vaccine hesitancy is a growing threat to global health security, and the WHO named vaccine hesitancy as one of the top 10 threats to global health in 2019 [34]. Despite the catastrophic impact of the pandemic and the enormous global effort to develop a vaccine as rapidly as possible, the COVID-19 vaccine may still be viewed with skepticism and hesitancy by some people on account of its unknown safety and long-term adverse effects. The results of the present study concerning the incidence of AEFI can be provided to HCWs to allowed them to better understand the actual experience of AEFI and the consequences of COVID-19 infection. Healthcare workers are vital in providing guidance and recommendations to patients and the wider community about vaccination in this global pandemic, which includes giving correct information on the risks and benefits of the vaccine, whether in developing or developed countries, and in urban or remote communities. In addition, reducing fear and enhancing understanding of the benefits, efficacy, and safety after COVID-19 vaccination is important for ensuring the continued well-being of HCWs.

This study has several limitations. Firstly, there was a much higher predominance of women than men among the study participants, consistent with the fact that nearly 75% of the world’s healthcare workers are women. Secondly, in China, the majority of vaccines involve inactivated COVID-19 vaccine (i.e., Vero cell), and this may explain the higher number of participants who received the Vero cell vaccine compared to the other two vaccines (i.e., CHO cell, Ad5 vectored). Thirdly, to avoid potential response fatigue, participants were not asked questions that focused on the severity level of every adverse effect. Therefore, the severity of each symptom was not assessed quantitatively in this study.

## 5. Conclusions

The present study has contributed to understanding the effect of COVID-19 vaccination on people in remote areas of China. The three COVID-19 vaccines that have been studied were all made in China and are broad-spectrum and safe. The potential risk of AEFI in the Ad5 vectored group may be higher than in the Vero cell group and CHO cell group, but none of the vaccines caused any serious or rare AEFI. AEFI were more common in women than men and more common in participants with a higher level of education. The use of this information to provide pre-and post-vaccination psychological and social support will increase the confidence of HCWs and the public to be vaccinated against COVID-19.

## Figures and Tables

**Figure 1 vaccines-10-01449-f001:**
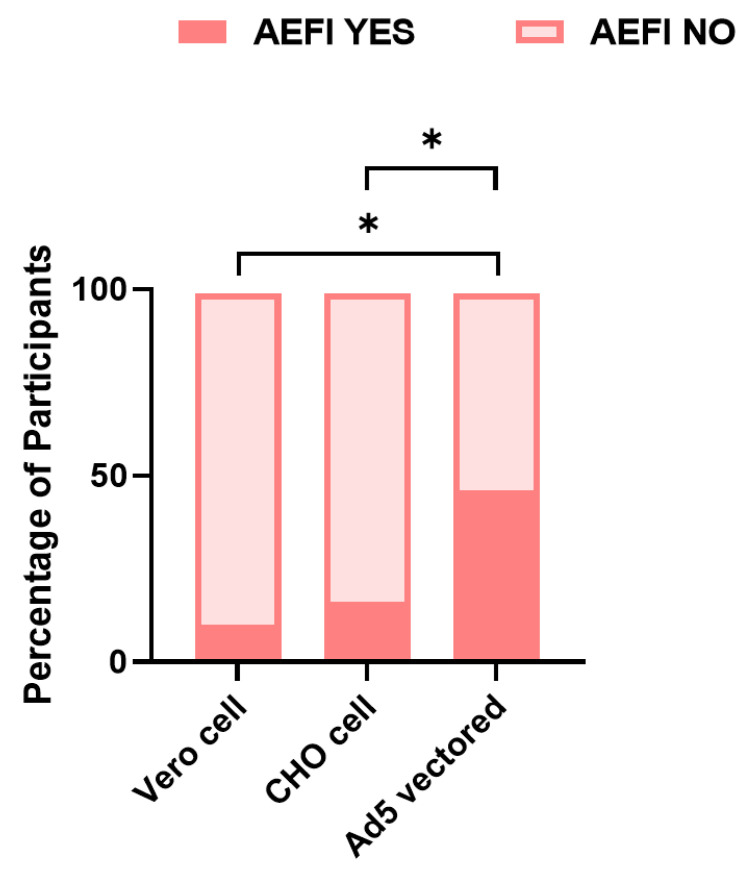
Statistical analysis performed using the χ^2^ test to compare the frequency of AEFI between 3 groups of vaccines. * *p*-value < 0.05 (*p*-value was adjusted by Bonferroni).

**Figure 2 vaccines-10-01449-f002:**
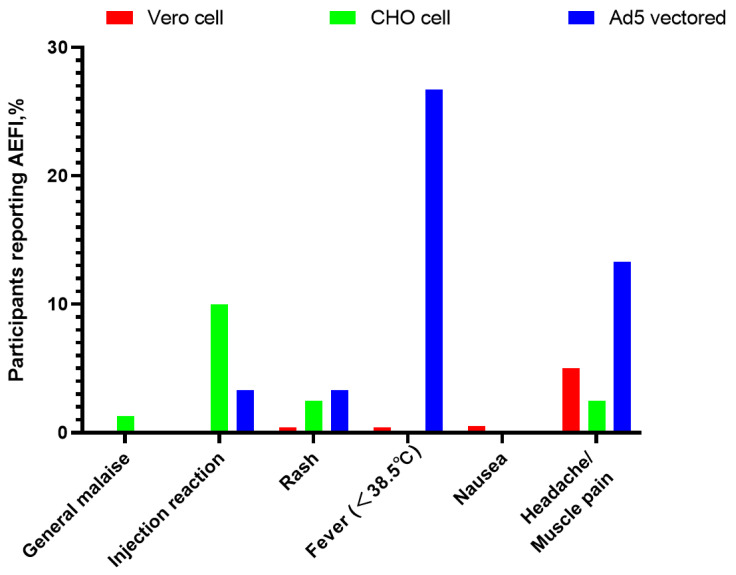
AEFI self-reported by participants within 3 days of receiving one of the three different types of COVID-19 vaccine.

**Table 1 vaccines-10-01449-t001:** Demographic data for the study participants who received one of the three types of COVID-19 vaccine.

	Vero Cell Group	Cho Cell Group	Ad5 Vectored Group	Total	Statistic	*p*-Value
Age group (in years)						
18–30	480 (60.5%)	65 (81.3%)	16 (53.3%)	561 (62.1%)	χ^2^ = 14.8	<0.022
31–40	166 (20.9%)	7 (8.8%)	8 (26.7%)	181 (20%)		
41–50	76 (9.6%)	5 (6.3%)	3 (10%)	84 (9.3%)		
51–60	72 (9.1%)	3 (3.8%)	3 (10%)	78 (8.6)		
Gender					χ^2^ = 21.9	<0.001
Female	562 (70.8%)	73 (91.3%)	28 (93.3%)	663 (73.3%)		
Male	232 (29.2%)	7 (8.8%)	2 (6.7%)	241 (26.7%)		
Ethnicity					χ^2^ = 1.3	0.969
Han	510 (64.2%)	54 (67.5%)	20 (66.7%)	584 (64.6%)		
Miao	72 (9.1%)	8 (10.0%)	3 (10.0%)	83 (9.2%)		
Buyi	58 (7.3)	5 (6.3)	1 (3.3%)	64 (7.1%)		
Others	154 (19.4%)	13 (16.3%)	6 (19.1%)	173 (19.1%)		
Level of education					χ^2^ = 24.4	<0.001
Bachelor’s degree or above	416 (52.4%)	19 (23.8%)	17 (56.7%)	452 (50%)		
Junior college or blew	378 (47.6%)	61 (76.3%)	13 (43.3%)	452 (50%)		

**Table 2 vaccines-10-01449-t002:** Medical attention received by study participants after receiving one of three types of COVID-19 vaccine.

	Vero Cell Group	Cho Cell Group	Ad5 Vectored Group	Total	Statistic	*p* Value
Regression of Symptom					Fisher’s exact test	<0.001
Symptomless	714 (90.3%)	67 (83.8%)	16 (53.3%)	797 (88.2%)		
Spontaneous remission	80 (10.1%)	11 (13.8%)	12 (40.0%)	103 (11.4%)		
Seek help from outpatient provider	0	2 (2.5%)	2 (6.7%)	4 (0.4)		

**Table 3 vaccines-10-01449-t003:** Risk factors for AEFI following administration of different COVID-19 vaccines in HCWs.

Variables	Univariate Analysis		Multivariate Analysis	
	OR (95% CI)	*p*	OR (95% CI)	*p*
Vero cell group	0.128 (0.060–0.272)	<0.001	0.141 (0.065–0.306)	<0.001
CHO cell group	0.222 (0.087–0.563)	0.002	0.279 (0.107–0.723)	0.009
Ad5 vectored group	Reference		Reference	
Female	2.428 (1.377–4.280)	0.002	2.093 (1.171–3.742)	0.013
male	Reference		Reference	
Bachelor’s degree or above	2.055 (1.348–3.134)	0.001	2.237 (1.434, 3.489)	<0.001
Junior college or blew	Reference		Reference	

Abbreviations: OR, odds ratio; CI, confidence interval; *p*, *p*-value.

## Data Availability

All data in the study are available from the corresponding author by request.

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
