# Peer review of "Analysis of Adverse Effects of COVID-19 Vaccines Experienced by Healthcare Workers at Guizhou Provincial Staff Hospital, China"

_vaccines, 2022, doi:10.3390/vaccines10091449_

Round 1
Reviewer 1 Report
- this is a survey, not a real study on registring AEFI
- Lacks primary/secondary hypotheses and ranking of hypotheses
- Lacks aims/ objective
- Despite EC decision, no informed consent was requested from volunteers
-why recovered from infection and older than 60 were excluded from the study?
-why the study group is NCW? Are there any previous reports on increased risk of AEFI in HCWs?
- Data not supporting conclusions on effectiveness and safety of these vaccines
Author Response
Dear reviewer,
we are very grateful for your comments on the manuscript. All your comments are very important, they are important for my paper writing and research. According to your advice, we amended the relevant part of the manuscript. Some of your questions were answered below.
Firstly, Our study was approved by the Institutional Review Board. The Institutional Review Board waived the necessity for written informed consent from each participant who participated in our study because the data we evaluated was de-identified. Furthermore, no personally identifiable information or human biospecimens were used. Therefore, our study did not violate the fundamental principles of EC.
Secondly, this was a web-based questionnaire on Adverse Events Following Immunization (AEFI) after COVID-19 vaccination among Healthcare workers (HCWs) in Guizhou Provincial Staff Hospital. This is a study conducted in the real world.
Thirdly, at the beginning of the COVID-19 epidemic, Guizhou Provincial Staff Hospital was designated by the local government as a specialized medical institution for the treatment of SARS-CoV-2 patients, but no HCWs were infected. Furthermore, in China, the working age of HCWs is 60 years old, so our survey was conducted among HCWs aged < 60, and this was the same age group as the publicly reported COVID-19 vaccine study.
Finally, Among the publicly available research reports, there are none in which the risk factors for AEFI after the use of different types of COVID-19 vaccines in HCWs are reported.
Hope to get your recognition. Thank you again for your advice. I hope I can learn more from you.
With best regards,
Yaying Li, MD, PhD

Reviewer 2 Report
This is an interesting study evaluating the occurrence of adverse effects against three types of COVID vaccines in professionals in a hospital in China.
The article reflects the effort made to carry out the project during the pandemic, which has been a difficult time to conduct studies. For this, my special recognition.
Despite the effort made, the article has important methodological limitations, which compromise the validity of the article.
From my point of view, the main limitation of the study is that, although it does not describe how the data collection procedure was carried out (which in itself is a limitation), everything indicates that the questionnaires were all filled out at the same time, after the entire vaccination period they considered had elapsed. In other words, the professionals did not record information on adverse effects in the days following the administration of the vaccine, but were asked afterwards, after a period of time, whether they had experienced these effects.
In addition to this problem, this study has other limitations, some of them relevant, such as the lack of informed consent, the interpretation of the statistical analysis or the description and discussion of the results, which should also be corrected. I have been adding my contributions, in comment format, in the attached document.
From my point of view, the important methodological limitations of the study condition the validity of the results and I consider that the article should not be accepted.
Kind regards

Author Response
Dear reviewer,
We greatly appreciate your comments on the manuscript. All your comments are important and important to my paper writing and research.
I have read your comments very carefully and conclude that articles should not be accepted because it they lacked a description of data collection procedures. We admitted that we did not include a description of data collection procedures in our manuscript although perhaps we should have done. This was for the sake of brevity rather than an error or omission. According to your advice, we have carefully amended the relevant part of the manuscript and I will answer your questions in the cover letter. Hope to get your recognition.
Please see the revised version of our manuscript form which is attached below. Thank you again for your advice. I hope I can learn more from you.
With best regards,
Yaying Li, MD, PhD

Reviewer 3 Report
This manuscript gives some input about the adverse effects that can be present in different types of COVID-19 vaccines.
The results are not a novelty and are in the range that could be expected.
The authors affirm that the information derived from the work can be useful regarding vaccination strategies. In view of the results, could the authors indicated what strategies could they consider?
Author Response
Dear reviewer,
Thank you for your suggestions. All of your suggestions are very important, and they have an important guiding significance for my thesis writing and scientific research work.According with your advice, we amended the relevant part in manuscript. Some of your questions were answered below.
The results are not a novelty but prove the safety of the vaccines. Moreover, In view of the results, we considered some strategies:
Firstly, In adults, women tend to exhibit a stronger immune system response to vaccines than men, which makes women more resistant to infectious diseases, but on the contrary, women have a higher incidence of autoimmune diseases. Therefore, Sex may be a critical factor in the design of future vaccine trials. The personalized vaccine customized for the changes in the gender immune diagram may provide greater protection for infectious and non -infectious targets.
Secondly, in addition to biological effects, differences in education level have been shown to influence the response to vaccines. We considered that although the medical staff are highly educated people, Guizhou province is a relatively remote ethnic region with an economy and cultural assets that lag behind that of more developed cities. COVID-19 led to the beginning of a worldwide pandemic, one of the most exposed groups to the virus and its psychosocial consequences is the healthcare workers. They are particularly vulnerable, given their risk of exposure to the virus, concern about infecting and caring for their loved ones, shortages of personal protective equipment, longer work hours, and involvement in emotionally and ethically fraught resource-allocation decisions. Unpleasant emotions can cause the body to generate very high stress levels which can disturb the functioning of the immune system, central nervous system and endocrine system.. Therefore, we proposed to pay attention to the psychological comfort and social care of HCWs, to provide psychological counseling for HCWs.
Please see the revised version of our manuscript form which is attached below. Hope to get your recognition. Thank you again for your advice. I hope I can learn more from you.
With best regards,
Yaying Li, MD, PhD
Round 2
Reviewer 2 Report
I believe that the review carried out has significantly improved the article. In this sense, I consider that the main improvement is related to the description of the procedure in which it is made clear that the recording of adverse effects was carried out simultaneously with the administration of the vaccine.
Please find attached some questions and comments on some aspects that continue to raise doubts in my mind.
Best regards.

Author Response
Dear reviewer,
We greatly appreciate your comments on the manuscript once again. All your comments are important and important to my paper writing and research.
According to your advice, we have carefully amended the relevant part of the manuscript and I will answer your questions in the following section. Hope to get your recognition.
Thank you again for your advice. I hope I can learn more from you.
With best regards,
Response to Reviewer 1 Comments
In the manuscript, we have summarized and answered the questions as follows:
1. I still don't understand this explanation of OR in logistic regression. What statistical books have you consulted to make this interpretation?
Answer: Thank you very much for this question, allowing us to resolve our doubts again. We have reviewed the literature in the “Odds Ratios—Current Best Practice".
"The odds ratio for men compared with women is the ratio of
the odds for men divided by the odds for women. In this case, the
unadjusted odds ratio is 1.03/0.74 = 1.39. Therefore, the odds for
men receiving industry payments are about 1.4 as large (40%
higher) compared with women. Note that the ratio of the odds is
different than the ratio of the probabilities because the probability
is not close to 0. The unadjusted ratio of the probabilities for men
and women (Tringale et al7 report each probability, but not the
ratio), the relative risk ratio, is 1.19 (0.51/0.43)."
This is a screenshot from the literature. For the explanation of the OR in the logical regression, the yellow highlight is what you want to mean, but what I want to mean is the blue highlight.
In our manuscript, for example, “the risk of AEFI in women was 1.093 (OR = 2.093, 95% CI: 1.171-3.742, P = 0.013) times higher than in men” that is “the odds for women occurring AEFI are about 2.1 (OR = 2.093, 95% CI: 1.171-3.742, P = 0.013) as large compared with men”
Thanks so much for your guidance. In our manuscript, we have revised in the Abstract and Results section.
2. I don't know if, as it is written, it is sufficiently explained that the questionnaire was sent the same day they received the vaccination.
Answer: We would set up a WeChat group for HCWs to send the questionnaire daily. Therefore, the HCWs who received one dose of the COVID-19 vaccine on the same day, used a mobile phone App to report the symptoms of AEFI that occurred within 3 days of vaccination.
3. In relation to the variables in the questionnaire, the text states:
The questionnaire included requests for information regarding sex, age, ethnicity, level of education, a history of COVID-19 infection, wishes in respect of COVID-19 vaccination, history of allergies to the vaccine, types of vaccinations, doses of vaccinations, symptoms of AEFI within 3 days of vaccination, and whether medical attention was required after administration of the vaccine.
The variable of the wish to receive the vaccine has been added, but then in the results nothing is said about it. I think it would be better not to include it.
Answer: We have added the relevant content in the Results section, paragraph 1. “Of the 1,000 HCWs who received the questionnaire, 4 have a weak desire to get vaccinated because of vaccine hesitation”.
4. It has to be very well explained in the text.
Answer: Thanks so much for your comment. We have deleted “the same day”.
5. It should be included whether the differences in medical care received, according to the vaccine received, are statistically significant.
Answer: According to your comments, we have revised. Please see Table 2.
6. In the text it is written:
There were significant differences in the incidence of AEFIs between the three groups (Figure 2), with values of 46.7%, 16.3% and 10.1% in the Ad5 vectored, CHO cell and Vero cell groups (c2 =38.7, P < 0.001), respectively (see Figure 1).
Information on the percentage of adverse effects is shown in figure 2, not in figure 1.
Answer: Thanks so much for your comments. In our manuscript, it has been revised.
7. Not included in the statistical analysis section.
Answer: Thanks so much for your comment. We have revised in the statistical analysis section.
